# Ice nucleators, bacterial cells and *Pseudomonas syringae* in precipitation at Jungfraujoch

Emiliano Stopelli[1], Franz Conen[1], Caroline Guilbaud[2], Jakob Zopfi[3], Christine Alewell[1], Cindy E. Morris[2]

[1]Environmental Geosciences, University of Basel, 4056 Basel, Switzerland
[2]INRA PACA, UR 0407 Plant Pathology Research Unit, 84143 Montfavet, France
[3]Acquatic and Stable Isotope Biogeochemistry, University of Basel, 4056 Basel, Switzerland

*Correspondence to*: Emiliano Stopelli (emiliano.stopelli@unibas.ch) and Franz Conen (franz.conen@unibas.ch)

**Abstract** Ice nucleation is a means by which the deposition of an airborne microorganism can be accelerated under favourable meteorological conditions. Analysis of 56 snow samples collected at the high altitude observatory Jungfraujoch (3580 m a.s.l.) revealed an order of magnitude larger dynamic range of ice nucleating particles active at -8 °C (INPs$_{-8}$) compared to the total number of bacterial cells (of which 60 % was on average alive). This indicates a shorter atmospheric residence time for INPs$_{-8}$. Furthermore, concentrations of INPs$_{-8}$ decreased much faster, with an increasing fraction of water precipitated from the air mass prior to sampling, than the number of total bacterial cells. Nevertheless, at high wind speeds (> 50 km h$^{-1}$) the ratio of INPs$_{-8}$ to total bacterial cells largely remained in a range between $10^{-2}$ to $10^{-3}$, independent of prior precipitation, likely because of recent injections of particles in regions upwind. Based on our field observations, we conclude that ice nucleators travel shorter legs of distance with the atmospheric water cycle than the majority of bacterial cells. A prominent ice nucleating bacterium, *Pseudomonas syringae*, has been previously supposed to benefit from this behaviour as a means to spread via the atmosphere and to colonise new host plants. Therefore, we targeted this bacterium with a selective cultivation approach. *P. syringae* was successfully isolated for the first time at such an altitude in 3 of 13 samples analysed. Colony-forming units of this species constituted a minor fraction ($10^{-4}$) of the numbers of INPs$_{-8}$ in these samples. Overall, our findings expand the geographic range of habitats where this bacterium has been found and corroborates theories on its robustness in the atmosphere and its propensity to spread to colonise new habitats.

## 1 Introduction

The nucleation of ice in clouds is a process of primary relevance both for the radiative budget of clouds and for the development of precipitation (Cantrell and Heymsfield, 2005; Mülmenstädt et al., 2015; Murray et al., 2012). Most ice nucleating particles (INPs) active at moderate supercooling in the atmosphere are of biological origin (Murray et al., 2012). *Pseudomonas syringae* was the first organism found to produce an ice nucleation active molecule (Maki et al., 1974) and to occur in clouds as potential biological INP (Sands et al., 1982). As it is also a plant pathogen it received and continues to receive attention from biologists in the perspective of improving the protection of crops from diseases (Lamichhane et al., 2014; Lamichhane et al., 2015). The combination of both roles, of an ice nucleator and of a plant epiphyte and pathogen, sparked the bioprecipitation hypothesis (Morris et al., 2014; Sands et al., 1982). Part of the hypothesis is the idea that ice nucleation activity contributes preferentially to the deposition of the organisms with this property helping them to return to plant surfaces where

they can proliferate (Morris et al., 2013a). In fact, favouring the growth of ice crystals could be a powerful and effective means for airborne microorganisms to reduce their residence time aloft. Previous research has illustrated that bacterial strains capable of nucleating ice can deposit rapidly under laboratory simulated cloud conditions, corresponding to a cold temperature regime and supersaturation (Amato et al., 2015). Modelling studies suggest that the atmospheric residence time of singularly airborne bacterial cells is highly reduced if such bacteria act as condensation and ice nuclei in clouds (Burrows et al., 2009). Furthermore, snowfall has been shown to enrich for the presence of ice nucleation active strains of *P. syringae* compared to a range of other environmental contexts including within clouds (Morris et al., 2008; Morris et al., 2013a). Therefore, more direct evidence from field observations on the selective deposition of ice nucleation active microorganisms with precipitation is wanting.

Here we explore, through analysis of snow samples collected at a high altitude station, whether the capability to induce the formation and growth of ice crystals makes a discernible difference to the atmospheric residence time of INPs compared to the majority of bacterial cells, which are not INPs. We have sought to isolate the prominent ice nucleation active bacterium *P. syringae* at high altitude and compare its abundance with numbers of $INPs_{-8}$. We have also carried out phylogenetic analysis in an attempt to identify potential sources of this bacterium in the atmosphere.

## 2 Materials and methods

### 2.1 Sampling and counting of INPs and bacteria

Fifty six snow samples were collected at the High Altitude Research Station Jungfraujoch (7°59'06'' E, 46°32'51'' N, 3580 m a.s.l., Switzerland) during 11 short sampling campaigns from March to September 2013 and May to October 2014. The Station was always inside precipitating clouds while collecting snow. Samples were collected with a 0.1 $m^2$ Teflon-coated tin for periods of 1.5 to 8 hours depending on precipitation intensity. The tin was rinsed several times with 70 % ethanol and sterile Milli-Q water between samples to avoid cross-contamination. Snow samples were melted at room temperature (about 16 °C) and adjusted to physiological conditions (9 ‰ NaCl) to prevent osmotic stress on cells and improve the detection of freezing events. They were analysed immediately on site for concentration of INPs active between -2 and -12 °C in a drop freeze assay with the LINDA device loaded with 52 Eppendorf Safelock tubes containing 100 µL of sample each, as described in Stopelli et al. (2014). For each sample, two filters were prepared for later analysis of bacterial number concentration. Twenty mL of sample were passed through the active area (glass vacuum filter funnels were equipped with an inlay to reduce the whole area of the filter into an active area of 6 mm diameter, to improve the possibility of counting enough cells per unit area of filter) of two 0.22 µm black polycarbonate membranes (Whatman). The filters were rinsed with 3 mL of sterile phosphate-buffered saline (PBS) and the staining agents were added: 10 µL SYBR green (100x) for total cell count, and 10 µL SYBR green and 10 µL propidium iodide (in Milli-Q water, 1‰ wt/wt) to facilitate counting cells that are alive (i.e with intact membranes) and dead or dying (with permeable membranes). After incubation in the dark for 10 min, the stains were filtered away and the filter columns rinsed with 3 mL of sterile water. Once dry, the filters were put on glass slides, 15 µL of antifading agent (5 mL PBS, 5 mL glycerol, 10 mg p-phenylendiamine) was added for preservation in the dark at -20 °C until analysis. Blanks for the determination of INPs and bacterial cells were

periodically prepared using the Milli-Q water used to rinse the tin as control sample. Blank counts were generally less than 10 % of sample counts. Filters were analysed with a fluorescence microscope (Leica DM2500) with a 100x ocular lens and an objective with 10x10 10 µm grids. Bacterial cells were recognised by shape and size. Each time, at least 10 fields and 300 cells were counted. In filters stained solely with SYBR green all cells were visible under blue light, while in filters treated with SYBR green and propidium iodide only living cells were counted.

## 2.2 Selective isolation and phylogenetic characterization of *Pseudomonas syringae*

Enough precipitation volume was available in 13 samples in a total of 15 samples collected in 2014 to assess the presence of culturable *P. syringae*. The method employed for the selective isolation of *P. syringae* and its metabolic characterisation is described in Monteil et al. (2014) and in Morris et al. (2008). In addition to the above described procedure, the snow collecting tin was sterilised by dry heat (150 °C) for 20 min. Samples were concentrated 500 times by filtration of about 1 litre of melted snow across sterile 0.22 µm pore size polycarbonate filters and subsequent resuspension of the particles by stirring the surface of the filter into 2 mL of filtered precipitation water. The concentrated samples were dilution-plated on KBC, King's medium B supplemented with boric acid, cephalexin and cycloheximide, to isolate and calculate the abundance of *P. syringae*. Plates were incubated at 20–25 °C for 3 days. Putative strains of *P. syringae* were purified on KB media (without the antibiotics and boric acid of KBC) where their production of fluorescent pigment could be observed under UV light (312 nm) and were then tested for the absence of arginine dihydrolase and for the absence of cytochrome C oxidase. Those that were negative for both the oxidase and arginine tests were suspended in sterile phosphate buffer and sent to the Plant Pathology Research Unit, INRA, Montfavet, France for molecular identification using PCR primers specific for *P. syringae*: Psy-F: 5'-ATGATCGGAGCGGACAAG-3'; Psy-R: 5'-GCTCTTGAGGCAAGCACT-3' (Guilbaud et al., 2016). The strains confirmed to be *P. syringae* were also tested for their ice nucleating activity. After 3 days of growth on KB, suspensions of pure colonies corresponding to $10^8$ cells mL$^{-1}$ in 9 ‰ NaCl were incubated 1 h in melting ice and subsequently tested for ice nucleation activity between 0 and -8 °C, with the freezing apparatus LINDA at 200 µL per tube (about $2\cdot10^7$ cells in each tube). The capacity of strains of *P. syringae* to induce a hypersensitive response (HR), indicative of the presence of a functional type III secretion system used in pathogenicity by the strains, was determined on tobacco plants (*Nicotiana tabacum*) by infiltrating bacterial suspensions of 48 h cultures at approximately $10^8$ cells mL$^{-1}$ into the leaves of the plant. After 24-48 h of incubation HR was revealed by the appearance of necrotic lesions at the site of infiltration.

Neighbour-joining phylogenetic trees of the strains of *P. syringae* isolated from Jungfraujoch were constructed on the basis of partial sequences of the citrate synthase housekeeping gene (*cts*) as previously described (Berge et al., 2014). Primers Cts-FP (forward): 5'-AGTTGATCATCGAGGGCGC(AT)GCC-3' and Cts-RP (reverse): 5'-TGATCGGTTTGATCTCGCACGG-3' were used for amplification and primer Cts-FS (forward): 5'-CCCGTCGAGCTGCCAAT(AT)TTGCTGA-3' for sequencing (Morris et al., 2010; Sarkar and Guttman, 2004). Analysis of partial *cts* gene sequences was performed as described previously using *P. syringae* reference strains (Berge et al., 2014), atmospheric strains (Amato et al., 2007; Joly et al., 2013; Vaitilingom et al., 2012) and the 24 strains from this study that were positive with the PCR specific for *P. syringae*. Alignment of sequences was made with DAMBE (version 5.6.8) and a Neighbour Joining tree was built with MEGA (version 5.05; Tamura et

al., 2011). Reported sequences are deposited in the GenBank Archive under the accession numbers GenBank
KY379248-KY379271.
**2.3 Environmental parameters and statistics**
Particles tend to be removed from the atmosphere by precipitation, specifically INPs (Stopelli et al., 2015).
According to this, it is relevant to know the fraction of water vapour lost as precipitation from an air mass prior
to sampling, which is calculated from the relative abundance of $^{18}$O and $^{16}$O in precipitation, expressed as $\delta^{18}$O.
Water molecules containing the stable isotope $^{18}$O have a greater propensity to condense, hence to precipitate,
than those containing the more abundant stable isotope $^{16}$O. Therefore, the relative abundance $^{18}$O:$^{16}$O in an air
mass decreases during precipitation. The fraction of water vapour lost can be easily calculated comparing the
isotopic composition of the water vapour in an air mass at the moment of its formation, assuming marine origin,
with the composition at the moment of sampling, according to Rayleigh's fractionation model (IAEA, 2001).
More details on this calculation are provided in Stopelli et al, 2015. Wind speed was measured at Jungfraujoch
by MeteoSwiss and data were stored as 10-minute averages.
Univariate and non-parametric statistics were carried out with PAST software version 2.17 (Hammer et al.,
15 2001).

**3 Results and Discussion**
**3.1 Variability of the concentrations of bacterial cells and INPs$_{-8}$**
Concentrations of total bacterial cells in the 56 samples collected over the course of 18 months (excluding the
winter period between 2013 and 2014) were mostly between $3.0 \cdot 10^3$ ($10^{th}$ percentile) and $3.9 \cdot 10^4$ mL$^{-1}$ ($90^{th}$
percentile), a range which is coherent with previous observations carried out on precipitation and cloud water
samples collected at several places around the world (Bauer et al., 2002; Joly et al., 2014; Šantl-Temkiv et al.,
2013; Sattler et al., 2001; Vaïtilingom et al., 2012). Concentrations of INPs active at -8 °C or warmer (INPs$_{-8}$)
(Fig. 1, data taken from Stopelli et al., 2016) presented a similar trend (Spearman's correlation coefficient $r_S$ =
0.45, p < 0.001) but with an order of magnitude larger dynamic range (DR, i.e. the ratio between the largest and
the smallest values) ($10^{th}$ percentile: 0.2 mL$^{-1}$; $90^{th}$ percentile: 68.1 mL$^{-1}$).
The DR of atmospheric particles or molecules is largely determined by their atmospheric residence time. In fact,
species with longer residence time have a higher chance to mix and integrate more sources and over wider areas.
Furthermore, a longer residence time leads to a higher background concentration of a species in the atmosphere.
Changing inputs due to changing source strength, or changing losses due to changing sink strength, only make
smaller differences to this larger pool. Hence, the relatively small DR of bacterial cells compared to that of INPs-
$_8$ suggests that the majority of bacterial cells has a longer residence time in the atmosphere than INPs$_{-8}$. The
shorter residence time of INPs$_{-8}$ is likely to be due to their capacity to catalyse ice formation and growth at -8 °C,
leading to their rapid deposition with the growing crystal. Such INPs$_{-8}$ include not only bacterial cells, but also
fungal spores, pollen, parts thereof, and soil particles associated with biological ice nucleation active fragments
(Conen et al., 2011; Fröhlich-Nowoisky et al., 2015; Hill et al., 2016; Joly et al., 2013; Morris et al., 2013b;
O'Sullivan et al., 2015, Pummer et al., 2015).
Surprisingly, the fraction of living cells among total bacterial cells was on average 0.6, with a small standard
deviation (0.1), despite the harsh environmental conditions such as low temperature (sometimes down to -25 °C)
and intense solar radiation. The fraction of living cells was neither related to the fraction of water lost prior to
sampling nor to wind speed, and had no relation with the number of $INPs_{-8}$. This finding does not exclude that
bacteria can constitute an important fraction of $INPs_{-8}$. Rather, it suggests that i) the ice nucleating capability can
be retained beyond cellular death under atmospheric conditions (Amato et al., 2015; Möhler et al., 2007) and/or
ii) it can be linked not only to entire cells but also to cellular fragments or molecules released from cells
(O'Sullivan et al., 2016; Pummer et al., 2015).
In an earlier study we had already found evidence for INPs being more efficiently deposited from precipitating
clouds than the majority of particles larger than 0.5 µm (Stopelli et al., 2015). This evidence was based on the
comparison of $INPs_{-8}$ in precipitation with that of particles in air. In this work, by comparing number
concentrations of both $INPs_{-8}$ and bacteria in precipitation, we corroborate with field data the hypothesis that the
ability to foster the formation and growth of ice crystals increases the chance of an airborne particle to be
deposited. With an increasing fraction of water precipitated from air masses prior to arrival at the observatory,
the number concentration of $INPs_{-8}$ decreased much faster than that of bacteria (Fig. 2). Therefore, a more rapid
loss from the atmosphere with precipitation is one factor contributing to the greater dynamic range of $INPs_{-8}$.
This opens the question whether we can find evidence also for the replenishment of their atmospheric
concentrations at shorter time scales.
**3.2 The influence of wind speed on $INPs_{-8}$ and bacteria**
Wind speed is an important factor associated with enhanced number concentrations of $INPs_{-8}$ in air masses at
Jungfraujoch (Stopelli et al., 2016). At high wind speeds (> 50 km h$^{-1}$) numbers of $INPs_{-8}$ were $10^{-2}$ to $10^{-3}$ times
the number of bacterial cells in precipitation, independent of prior precipitation (Fig. 3, red symbols). At lower
wind speeds, the number of $INPs_{-8}$ decreased much more rapidly than the number of bacterial cells (blue
symbols). We interpret this observation in the following way. High wind speeds at Jungfraujoch could be
associated also with high wind speeds in source regions upwind (e.g.: Swiss Plateau, Po Valley) which promote
the aerosolisation of $INPs_{-8}$ and bacterial cells (Fig. 2; Lindemann and Upper, 1985). The atmospheric residence
time of total bacterial cells is longer than that of $INPs_{-8}$. Therefore, their background number concentration
would be relatively large and stable and not changed much by the additional cells aerosolised at high wind
speeds in the region. The shorter atmospheric lifetime of $INPs_{-8}$ means their background number concentration
tends to be smaller, relative to the emission strength of these particles. Consequently, numbers of $INPs_{-8}$ increase
more substantially above background values at high wind speed, than in the case of bacterial cells. Hence, the
ratio of $INPs_{-8}$ to bacterial cells remains large, even when a substantial fraction of water had already precipitated
from the air mass prior to sampling. At lower wind speeds, concentrations of $INPs_{-8}$ are not restored from
regional sources and we see more clearly the effect of a preferential washout of $INPs_{-8}$ relative to bacterial cells
(blue symbols in Fig. 3). The median ratio of $INPs_{-8}$ to bacterial cells we observed at Jungfraujoch was $6.6 \cdot 10^{-4}$,
very close to what Joly et al. (2014) had found over the course of a year in cloud water at Puy de Dôme ($5.5 \cdot 10^{-4}$).

### 3.3 Abundance and diversity of the prominent ice nucleating bacterium *P. syringae*

*Pseudomonas syringae* was successfully detected in three of the 13 samples analysed for the presence of this
bacterium. These samples had over 1000 bacterial cells $mL^{-1}$ and more than 10 $INP_{-8}$ $mL^{-1}$ and were from
clouds that had precipitated less than 70% of their water vapor prior to sampling (Fig. 2). Two-thirds of the
strains (16/24), after culture in the laboratory, produced at least 1 cell in a suspension of $2 \cdot 10^{7}$ cells that was ice
nucleation active at temperatures warmer than -8 °C. The freezing onset temperature of all ice nucleation active
strains was warmer than -5 °C and was -2.1 °C for the most active strain (Table 1). The precipitation samples
from which these strains were isolated were characterised by a relatively warmer onset of freezing (median onset
temperature of -5.0 °C for samples with *P. syringae* vs. -6.8 °C for those without *P. syringae*, p = 0.02, Mann-
Whitney test). Although *P. syringae* was found in the samples with the largest numbers of $INP_{-8}$ and with the
warmest onset freezing temperature, only 2, 4, and 45 colony-forming units (i.e. culturable cells) of *P. syringae*
were present per litre of sample. This corresponds to 2 orders of magnitude less than what has been found in
snowfall at lower altitudes (Monteil et al., 2014). We suspect that this can be due both to the preferential removal
of *P. syringae* with precipitation as soon as it develops at lower altitudes and to the larger distance of
Jungfraujoch from the sources where *P. syringae* entered into air masses. This possibly leads to longer exposure
of *P. syringae* to UV radiation and desiccation, reducing its culturability when collected at higher altitude. The
largest number concentration of colony-forming units of *P. syringae* we found in snow water was $10^{-4}$ times the
number of $INPs_{-8}$ in the same sample.
This is the first time this bacterium has been isolated at such altitudes (3580 m a.s.l.), and therefore this result
expands the established limits for *P. syringae*'s dissemination and survival under atmospheric conditions.
Sequencing of the *cts* gene for phylogenetic analyses was conducted for all 24 strains of *P. syringae* from culture
plates from the precipitation samples (Table 1) to obtain insight into the possible origin of these strains. For all
dates where *P. syringae* was isolated, the strains in each precipitation event were genetically diverse and
represented a broad range of known phylogenetic groups (Table 1, Supplemental Fig. 1). This high diversity
suggests that the process of entry of *P. syringae* into air masses moving up to Jungfraujoch involves either
multiple events from a wide range of sources along the trajectory or a few entry events from common sources
that harbour a high diversity of *P. syringae* that can readily be wafted into the air. Leaf litter, for instance, is one
substrate that could have been a source of a high diversity of *P. syringae*, since it contains a high density and
high genetic diversity of *P. syringae,* many of which are ice nucleation active, no matter the geographic origin
and trajectory of the air masses (Berge et al., 2014; Monteil et al., 2012).
All but 3 of the 24 strains had functional type III secretion systems (hypersensitivity test). This suggests that they
had potential to cause plant disease on some crops and illustrates the potential extent of spread of diseases
caused by this pathogen.

### 4 Conclusions

Based on observations of INPs$_{-8}$ and bacterial cells in 56 precipitation samples collected at Jungfraujoch we
have shown that ice nucleation activity at -8 °C and warmer temperatures contributes to shorter atmospheric
residence times due to a greater probability for wet deposition. For bacterial cells that are disseminated via the
atmosphere, this property is advantageous because, by enhancing deposition, the bacteria reduce the time of
exposure of their cells to hostile conditions such as UV radiation, extreme cold temperatures and to drying. The
decrease of INPs relative to bacterial cells with precipitation can be delayed by high wind speed, which promotes
the continuous uptake, mixing and transport of INPs over longer distances. Over half of the bacterial cells in
precipitation that fell on Jungfraujoch were viable and among those *P. syringae* could be cultured from 3 of 13
samples. But their concentration in precipitation was less than 50 culturable cells per litre of snow meltwater,
about $10^4$ times less than the concentration of INPs$_{-8}$. Therefore, culturable *P. syringae* appeared as a minor
component of the ensemble of INPs$_{-8}$ collected in precipitation at Jungfraujoch. Nevertheless, the presence in
snowfall on Jungfraujoch of a bacterium such as *P. syringae* sheds new light on the possibilities for this
bacterium to survive journeys through the atmosphere and colonise new plants and new habitats. At the same
time, it opens exciting research perspectives. For *P. syringae* there are a wide array of techniques for the
characterisation of its phenotypic and genotypic variability and banks of strains and genomic data related to
habitats and geographic origin. This makes *P. syringae* a powerful model for attempting to identify specific
sources of INPs in precipitating air masses and for better defining the extent of their trajectories.
**5 Data availability**
The data set for this paper is publicly available as Table in the Supplement.
**6 Author contributions**
Emiliano Stopelli carried out the field measurements at Jungfraujoch on the abundance of INPs, bacterial cells
and on the isolation of *P. syringae*, and together with Franz Conen analysed the data. Caroline Guilbaud
provided fundamental support in the isolation of *P. syringae* and did the phylogenetic analysis on the isolated
strains. Jakob Zopfi provided great help with the set up of epifluorescence microscopy technique. Christine
Alewell and Cindy E. Morris provided strong conceptual frameworks. Emiliano Stopelli wrote the manuscript
together with Franz Conen, with important contributions from all other co-authors.
The authors declare that they have no conflict of interest.
**7 Acknowledgements**
We thank the International Foundation for High Alpine Research Station Jungfraujoch and Gornergrat (HFSJG)
for making it possible for us to conduct our measurements at Jungfraujoch; Urs and Maria Otz, as well as Joan
and Martin Fisher are acknowledged for their support on site. Corinne Baudinot measured the abundance of INPs
in snow samples during the second year of observations. Dr Thomas Kuhn and Mark Rollog analysed the stable
isotope ratio in our snow water samples. We thank MeteoSwiss for providing data on meteorology at
Jungfraujoch. The work described here was funded by the Swiss National Science Foundation (SNF) through
grant no 200021_140228 and 200020_159194.

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

**Table 1.** Diversity of *Pseudomonas syringae* strains from fresh snow samples collected at Jungfraujoch.

| Strain | Date | Phylogroup[1] | INA[2] | HR[3] |
|---|---|---|---|---|
| JFJ-0007 | 22 May 2014 | 2 | -4.2 | + |
| JFJ-0008 | 22-23 May 2014 | 3 | - | - |
| JFJ-0011 | 22-23 May 2014 | 5 | -3.7 | + |
| JFJ-0039 | 21-22 Oct 2014 | 1 | - | + |
| JFJ-0045 | 21-22 Oct 2014 | 1 | - | + |
| JFJ-0048 | 21-22 Oct 2014 | 1 | - | + |
| JFJ-0061 | 21-22 Oct 2014 | 1 | - | + |
| JFJ-0043 | 21-22 Oct 2014 | 2 | -2.6 | + |
| JFJ-0047 | 21-22 Oct 2014 | 2 | - | - |
| JFJ-0050 | 21-22 Oct 2014 | 2 | -3.6 | + |
| JFJ-0052 | 21-22 Oct 2014 | 2 | -2.7 | + |
| JFJ-0054 | 21-22 Oct 2014 | 2 | -2.2 | + |
| JFJ-0055 | 21-22 Oct 2014 | 2 | -2.2 | + |
| JFJ-0058 | 21-22 Oct 2014 | 4 | - | + |
| JFJ-0056 | 21-22 Oct 2014 | 7 | -2.9 | + |
| JFJ-0059 | 21-22 Oct 2014 | 7 | -4.9 | + |
| JFJ-0040 | 21-22 Oct 2014 | 10 | -4.2 | + |
| JFJ-0044 | 21-22 Oct 2014 | 10 | -2.1 | + |
| JFJ-0049 | 21-22 Oct 2014 | 10 | -2.8 | + |
| JFJ-0051 | 21-22 Oct 2014 | 10 | -2.9 | + |
| JFJ-0053 | 21-22 Oct 2014 | 10 | -4.0 | + |
| JFJ-0060 | 21-22 Oct 2014 | 10 | -2.8 | + |
| JFJ-0046 | 21-22 Oct 2014 | 13 | - | - |
| JFJ-0057 | 21-22 Oct 2014 | 13 | - | + |

[1]Full details of the phylogenetic situation of these strains compared to a range of reference strains is presented in the Supplemental Figure 1.

[2]INA refers to ice nucleation activity of suspensions of $2 \cdot 10^7$ cells. The reported values are the freezing onset temperature.

[3]Capacity to induce a hypersensitive reaction (HR) in tobacco indicative of the presence of a functional type III secretion system that is one of the fundamental traits usually required for pathogenicity of *P. syringae* to plants.

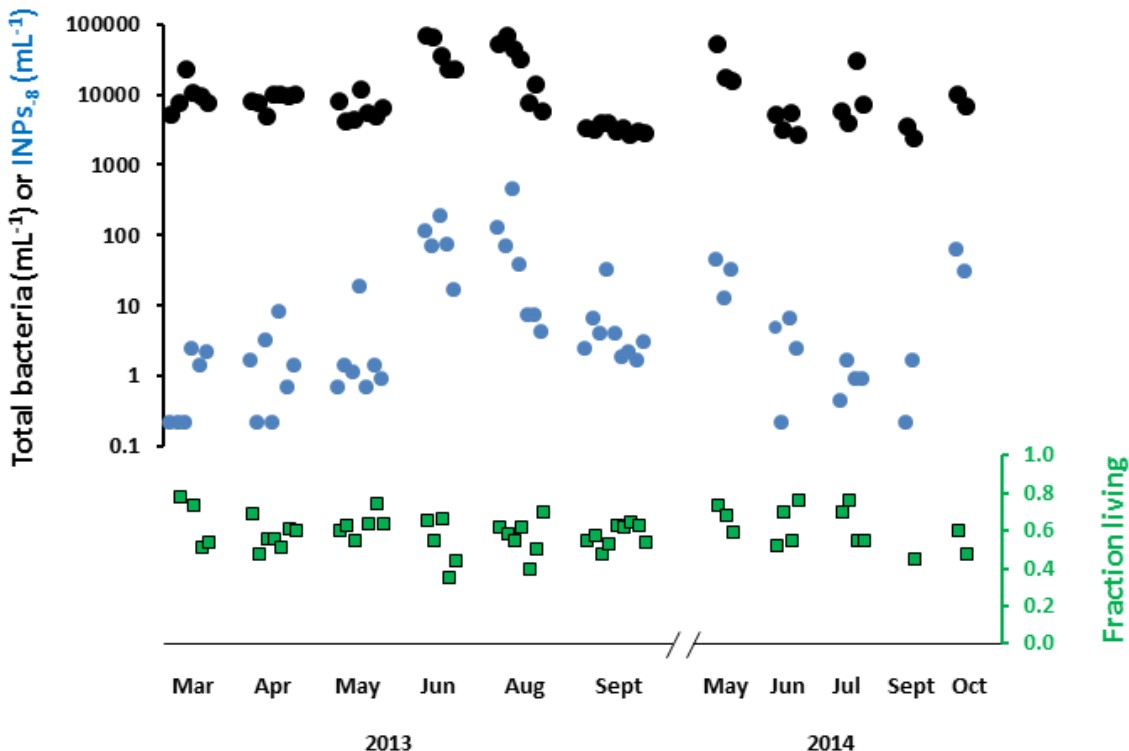

**Figure 1.** Number of total bacterial cells (black dots), fraction of living cells (green squares) and ice nucleating

particles active at -8 °C (INPs$_{-8}$, blue, from Stopelli et al., 2016) in precipitation samples collected at the high

altitude research station Jungfraujoch (3580 m a.s.l.) during 11 sampling campaigns between March 2013 and

October 2014.

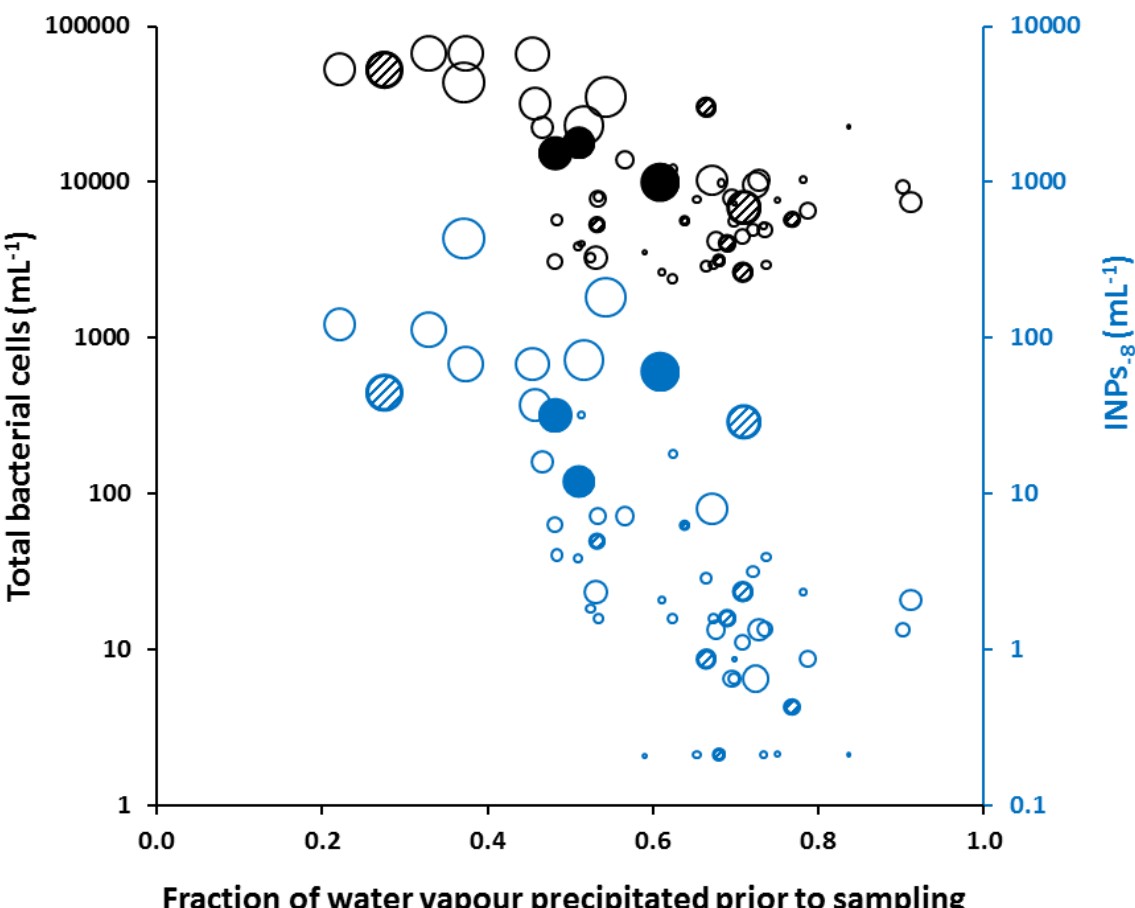

**Figure 2.** Number of bacterial cells (black) and INPs$_{-8}$ (blue, from Stopelli et al., 2016) versus the fraction of precipitation lost from the air mass prior to arrival and sampling at the observatory. The width of the symbols is proportional to wind speed (minimum = 2 km h$^{-1}$, maximum = 89 km h$^{-1}$). Patterned symbols represent the 10 fresh snow samples in which *P. syringae* was searched for, but not found. The three full symbols represent the 3 samples where culturable *P. syringae* was found.

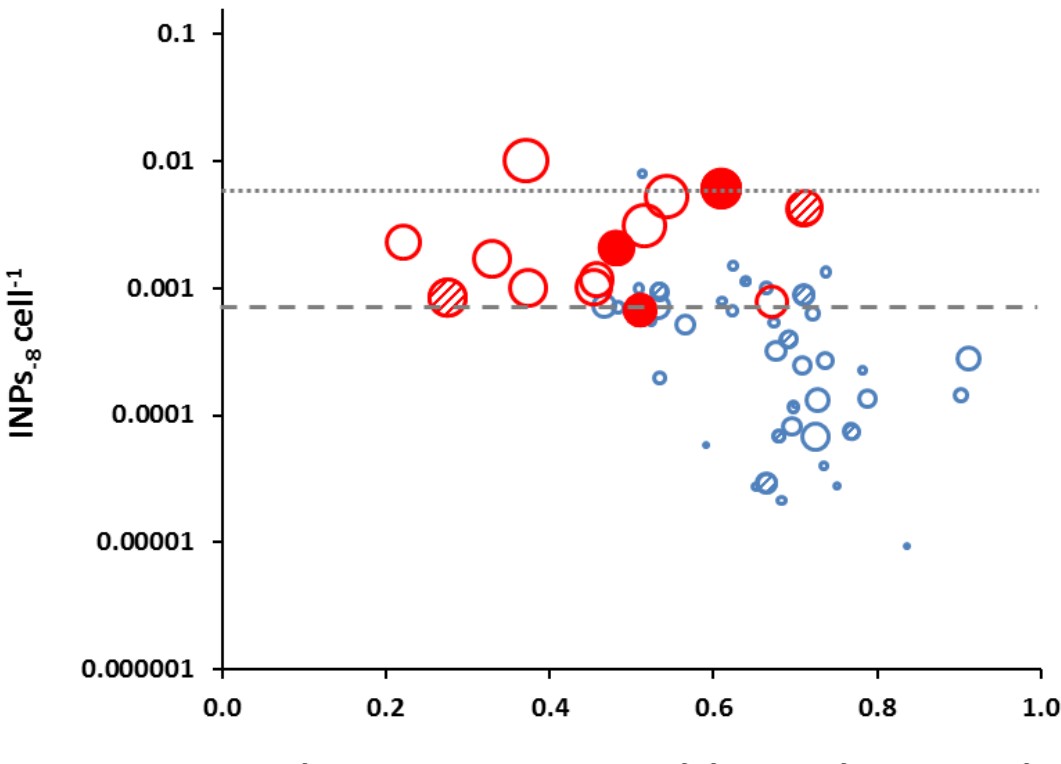

**Figure 3.** Ratio of INPs$_{-8}$ to bacterial cells in precipitation samples collected at Jungfraujoch (3580 m a.s.l.). The width of symbols is proportional to wind speed (minimum = 2 km h$^{-1}$, maximum = 89 km h$^{-1}$). Red symbols correspond to values with average wind speeds > 50 km h$^{-1}$ during samplings and blue symbols correspond to data with lower wind speeds. Patterned symbols represent the 10 samples in which *P. syringae* was searched for, but not found. The three full symbols represent samples where culturable *P. syringae* was found. Dashed and dotted lines indicate median and maximum ratios observed in cloud water collected over the course of a year at Puy de Dôme (1465 m a.s.l.), 350 km west of Jungfraujoch (data from Joly et al., 2014).

