# Peer review of "Ice nucleators, bacterial cells and Pseudomonas syringae in"

_Biogeosciences, 2016_

## Referee Comment (RC1) · Anonymous Referee #1 · 16 Dec 2016

Summary:

This manuscript presents the analysis of 56 fresh fallen snow sampled at Jungfraujoch for which concentrations of Ice Nucleating Particles (INPs) and bacteria have been determined and the fraction of water vapour lost by the air mass prior the sampling has been assessed by oxygen isotopic ratio. Based on the difference of dynamic ranges between INPs and bacteria concentrations, authors propose that INPs are more rapidly deposited from the atmosphere than bacteria. They confirmed it by a faster decrease of INPs than bacterial cells when air masses are submitted to larger losses of water vapour. A second part of the manuscript emphasizes the wind as a determining factor in the concentration of INPs. Finally, the authors isolated Pseudomonas syringae from the snow and compared them to strains isolated from a large panel of environments.

[Figure]

General comments:

The manuscript is clear, well-written, and in perfect adequacy with the scope of the journal. It brings arguments to elucidate what happens in clouds, and more generally in the atmosphere, where a lot of processes concerning microorganisms are supposed but not clearly demonstrated. Still, I have some concerns/questions about different parts of the manuscript:

1. I regret the structure of this paper in two parts: the first concerning INPs and bacteria persistence in the atmosphere and the second part concerning P. syringae and its belonging to specific phylogenetic groups. These two parts although comprehensible in atmospheric microbiology form a disconnected content.

2. In the first Results section, authors determine "dynamic ranges" to evaluate atmospheric residence times and support this citing "Fig 2 in Griffiths et al., 2014". First, I do not believe this citation is relevant in this context. Second, I wonder how efficient this index would be for example in the case of a particle with a short residence time and important emission sources all around the sampling site?

3. Authors measured total bacteria concentration and INPs-8. Is there a statistical correlation between these two parameters?

4. Title should clearly exposed authors were interested in INPs active for temperatures warmer than -8°C (and as much as possible an insight of the work on P. syringae).

Other comments:

P1 Line 22: It sounds like it is the first time P. syringae is found in snow: it is the first time at such an altitude, please precise it.

P1 Line 27: Hemsfield (missing "i")

P1 Line 30: Modify this sentence: Sands et al. (1982) did not demonstrate that P. syringae actually act as an INP in clouds

P2 Line 23 : why did you adjust samples to physiological conditions? What was the range of temperatures tested?

P2 Line 25: Are you sure the filter active area is 6 mm diameter?

P3 Line 10: How long were the plates incubated?

P3 Line 37: I would appreciate a brief description of this method.

P4 Line 5: Is there a scientific reason you excluded the winter period for sampling?

P4 Line 20: This sentence is ambiguous, please reformulate.

P4 Line 10: I would remove "seasonal" as you sampled only 7 months of the year. Furthermore, bacterial concentration in May 2014 is higher or equivalent to those in July/August.

P5 Line 16: Replace "is" with "in"

---

## Referee Comment (RC2) · Anonymous Referee #2 · 21 Dec 2016

General comments

This study represents the results of INP measurements and bacteria cell counting of 56 snow samples collected at Jungfraujoch over an 18-month sampling period. They found a larger dynamic range of INP active at -8°C compared to the number of bacterial cells and a high fraction of living bacteria cells. Correlations with water vapor loss prior sampling show that INP have shorter atmospheric residence times than bacterial cells, in particular at low wind speeds. Furthermore, 24 strains of Pseudomonas syringae were isolated from selected samples and phylogenetically characterized.

The manuscript is well written and the results are clear and well presented.

Specific comments

1. I suggest renaming the section 3 from "results" to "results and discussion" as there is no separate discussion section.

2. When discussing the higher concentration dynamics of the INP compared to bacteria the authors mention changing source and sink strength of INP and bring a one short statement about "other particles carrying a biological INP" on P4 Line20. This discussion seems very general and should be extended. Other possible biological INP sources (at least fungi) should be specifically mentioned and more references should be cited. This would also help to explain the INP-8 concentrations at JFJ as P. syringae was found only in three out of 13 tested samples. Given the high diversity of P. syringae strains as found in those samples and an average of 60% of total living bacteria cells in the samples one could actually expect to find P. syringae in more samples. Preferential removal of P. syringae with precipitation and loss of culturability during atmospheric transport as stated in the MS seems possible, but possible sources of non-P. syringae INP active at -8°C should be discussed in more detail.

3. Some more information about the sampling conditions would be helpful as the station at JFJ is not always within clouds. Did the authors only collect precipitation/snow when the station was within clouds? And if not, did they found differences in the number of INP and bacterial cells in samples collected in a cloud compared to samples collected below a cloud? Why was there no sampling in winter?

Other comments/typos:

P2 Line 23: Although the math is correct here, I think it is better to consistently use either 0.9 % (as on P3 Line 18) or 9 ‰ NaCl.

P2 Line 23: Please provide some basic information here. How many droplets did you measure? What was the volume of the droplets? What kind of controls were measured?

Table 1 caption: I suggest to change the beginning to "Diversity of Pseudomonas syringae strains. . ."

[Figure]

---

## Author Comment (AC2) · 13 Feb 2017

We thank the Referees for their constructive comments. Our detailed responses have been submitted in the attached pdf.

Please also note the supplement to this comment:
http://www.biogeosciences-discuss.net/bg-2016-496/bg-2016-496-AC2-supplement.pdf

---

## Author Response (AR1)

We thank the Referees for their constructive suggestions, which were of great help to improve the quality of the work and to present the results in a clearer way. Here is a point-by-point reply to their proposed corrections. We merged the replies into a unique file to improve its clarity and the level of detail of the answers.

*Referees' comments are in italic font,* authors' responses in blue normal font, changes in the manuscript are in blue underlined and refer to the version of the document published on BGD. To facilitate the discussion, we have numbered the comments.

**Referee #1**

This manuscript presents the analysis of 56 fresh fallen snow sampled at Jungfraujoch for which concentrations of Ice Nucleating Particles (INPs) and bacteria have been determined and the fraction of water vapour lost by the air mass prior the sampling has been assessed by oxygen isotopic ratio. Based on the difference of dynamic ranges between INPs and bacteria concentrations, authors propose that INPs are more rapidly deposited from the atmosphere than bacteria. They confirmed it by a faster decrease of INPs than bacterial cells when air masses are submitted to larger losses of water vapour. A second part of the manuscript emphasizes the wind as a determining factor in the concentration of INPs. Finally, the authors isolated Pseudomonas syringae from the snow and compared them to strains isolated from a large panel of environments.

General comments: The manuscript is clear, well-written, and in perfect adequacy with the scope of the journal. It brings arguments to elucidate what happens in clouds, and more generally in the atmosphere, where a lot of processes concerning microorganisms are supposed but not clearly demonstrated. Still, I have some concerns/questions about different parts of the manuscript:

(1) I regret the structure of this paper in two parts: the first concerning INPs and bacteria persistence in the atmosphere and the second part concerning P. syringae and its belonging to specific phylogenetic groups. These two parts although comprehensible in atmospheric microbiology form a disconnected content.

To improve the level of connection of the contents of the manuscript, we made some adjustments in the abstract at page 1 line 20, which now reads as: "A prominent ice nucleating bacterium, *Pseudomonas syringae*, has been previously supposed to benefit from this behaviour as a means to spread via the atmosphere and to colonise new host plants. Therefore, we targeted this bacterium with a selective cultivation approach. *P. syringae* was successfully isolated for the first time at such an altitude in 3 of 13 samples analysed". Paragraph 3.3 at page 5 line 22 was renamed in "Abundance and diversity of the prominent ice nucleating bacterium *P. syringae*". Finally, according to the comment 4 Referee #1 and the Editor's comment, the title of the manuscript has been changed into: "Ice nucleators, bacterial cells and *Pseudomonas syringae* in precipitation at Jungfraujoch".

(2) In the first Results section, authors determine "dynamic ranges" to evaluate atmospheric residence times and support this citing "Fig 2 in Griffiths et al., 2014". First, I do not believe this citation is relevant in this context.

We removed this sentence and the reference.

Second, I wonder how efficient this index would be for example in the case of a particle with a short residence time and important emission sources all around the sampling site?

In this case, the number concentrations of this type of particle would highly vary with the changing strength of the drivers of its emission (wind speed, radiation, surface wetness, etc.). At the same time, its background from upwind the source would be relatively small (short residence time, large local source). Consequently, we would also expect to see a large dynamic range in the concentrations of this particle type.

Nevertheless, at Jungfraujoch, we are about 1000 m above the nearest alpine meadows which could provide INPs. Furthermore, we do not expect the difference between alpine meadows and other

sources of bacteria and INPs, like lower lying meadows, forests and crops to be so large that this difference would be an alternative explanation for our observations.

In case glaciers are addresses as local sources of INPs, we have indications to exclude their influence in our measurements. We collected precipitation horizontally on the terrace of the observatory to maximize the recovery of falling snowflakes instead of particles floating. Lloyd et al., 2015 indicate that snow blown from the local sources around the observatory can increase the numbers of airborne ice particles (IPs) specifically under low wind speed. In our case, though, we measure INPs by immersion freezing and not IPs or secondary IPs, moreover we observe the opposite behavior with more INPs in association with high wind speed. Furthermore, if INPs are picked up from the surface of the glacier, these should have been freshly deposited on the snow surfaces around the station by the same air masses originating precipitation above the station, which makes them not much different from the one we collect on the terrace of the observatory.

**(3) Authors measured total bacteria concentration and $INPs_{-8}$ . Is there a statistical correlation between these two parameters?**

The two parameters are correlated, with Spearman's correlation coefficient  $r_s = 0.45$  and a statistical significance < 0.001. This justifies their similar trend, therefore at page 4 line 10 the sentence has been changed into: "Concentrations of INPs active at -8 °C or warmer (INPs-8) (Fig. 1, data taken from Stopelli et al., 2016) presented a similar trend (Spearman's correlation coefficient  $r_s = 0.45$ , p < 0.001) but with an order of magnitude larger dynamic range..."

At a closer look, though, this correlation is principally due to just a small set of samples where the total number of bacteria is larger than 1000 mL-1. Excluding those data, no correlation is present anymore. Therefore, we prefer to consider such correlation with care and to focus in this manuscript on the reasons why the numbers of bacteria do not vary as largely as  $INPs_{-8}$  do.

**(4) Title should clearly expose authors were interested in INPs active for temperatures warmer than - $8^{\circ}C$ (and as much as possible an insight of the work on P. syringae).**

The title was changed into: "Ice nucleators, bacterial cells and *Pseudomonas syringae* in precipitation at Jungfraujoch". Considering the Editor's comment, we decided not to provide too much level of detail in the title, also because the reference to INPs active at temperatures warmer than -8 °C is immediately visible in the first lines of the abstract. The title has been changed accordingly also in the Supplement.

**Other comments:**

**(5) P1 Line 22: It sounds like it is the first time P. syringae is found in snow: it is the first time at such an altitude, please precise it.**

This part of the abstract now reads as: "*P. syringae* was successfully isolated for the first time at such an altitude in 3 of 13 samples analysed".

**(6) P1 Line 27: Hemsfield (missing "i")**

**Corrected**

**(7) P1 Line 30: Modify this sentence: Sands et al. (1982) did not demonstrate that P. syringae actually act as an INP in clouds**

In that study, *P. syringae* was isolated from clouds and the ice nucleation activity of the strains tested under laboratory conditions. Therefore, the sentence at page 1 line 30 has been changed and now reads

as: "*Pseudomonas syringae* was the first organism found to produce an ice nucleation active molecule (Maki et al., 1974) and to occur in clouds as potential biological INP (Sands et al., 1982)".

**(8) P2 Line 23: why did you adjust samples to physiological conditions? What was the range of temperatures tested?**

The addition of small amounts of salts improves the detection of freezing events using our device LINDA. Furthermore, the final concentration 9 ‰ of NaCl should avoid osmotic stress to living cells, preventing breaking of cells and potentially multiplication of INPs of biological origin in samples. As discussed in Stopelli et al., 2014, the addition of this amount of salt does not depress the temperature of freezing of a sample. We analysed the temperatures between -2 and -12 °C. Considering these observations, and comment 5 Referee #2, the title of the section 2.1 now reads as: "2.1 Sampling and counting of INPs and bacteria"; while the paragraph has been modified into: "Snow samples were melted at room temperature (about 16 °C) and adjusted to physiological conditions (9 ‰ NaCl) to prevent osmotic stress on cells and improve the detection of freezing events. They were analysed immediately on site for concentration of INPs active between -2 and -12 °C in a drop freeze assay with the LINDA device loaded with 52 Eppendorf Safelock tubes containing 100  $\mu$ L of sample each, as described in Stopelli et al. (2014). For each sample, two filters were prepared for later analysis of bacterial number concentration."

**(9) P2 Line 25: Are you sure the filter active area is 6 mm diameter?**

Yes, we mounted an inlay of 6 mm diameter on the glass funnels used to filter the sample, in order to concentrate cells on a smaller active area, thus improving the quality of cell counts per unit area of filter. This detail has been inserted in the sentence, which now reads as: "Twenty mL of sample were passed through the active area (glass vacuum filter funnels were equipped with an inlay to reduce the whole area of the filter into an active area of 6 mm diameter, to improve the possibility of counting enough cells per unit area of filter) of two 0.22 µm black polycarbonate membranes (Whatman)".

**(10) P3 Line 10: How long were the plates incubated?**

We changed the sentence into: "Plates were incubated at 20-25 °C for 3 days."

**(11) P3 Line 37: I would appreciate a brief description of this method.**

Page 3 line 37 has been enriched by a brief description of this method and now reads as: "Particles tend to be removed from the atmosphere by precipitation, specifically INPs (Stopelli et al., 2015). According to this, it is relevant to know the fraction of water vapour lost as precipitation from an air mass prior to sampling, which is calculated from the relative abundance of 18O and 16O in precipitation, expressed as  $\delta^{18}$ O. Water molecules containing the stable isotope 18O have a greater propensity to condense, hence to precipitate, than those containing the more abundant stable isotope 16O. Therefore, the relative abundance 18O:16O in an air mass decreases during precipitation. The fraction of water vapour lost can be easily calculated comparing the isotopic composition of the water vapour in an air mass at the moment of its formation, assuming marine origin, with the composition at the moment of sampling, according to Rayleigh's fractionation model (IAEA, 2001). More details on this calculation are provided in Stopelli et al., 2015". Coherently, the reference to IAEA, 2001 has been added to the reference list.

**(12) P4 Line 5: Is there a scientific reason you excluded the winter period for sampling?**

Winter months have been sampled for the abundance of INPs-8 between the years 2012 and 2013 and the results are fully reported in Stopelli et al., 2016. Measurements of bacterial cells have been introduced from March 2013 on and in this paper we report only samples on which both measurements have been carried out. For the second year of observations (2014), we preferred to focus on the period

stretching from Spring to Autumn since we were expecting to sufficiently cover both small and large numbers of INPs and bacterial cells, whereas in winter numbers were suspected to be only very small as Jungfraujoch is surrounded by free tropospheric air for most of the time. Sampling revealed large variability of INPs not only across a year but also within sampling campaigns and in summer months. For a detailed discussion on the role of environmental parameters on the abundance of INPs please refer to Stopelli et al., 2016. Concerning bacterial cells, such large variability in the abundance does not occur also for Summer months and we examine possible causes of this different behaviour in this manuscript.

**(13) P4 Line 20: This sentence is ambiguous, please reformulate.**

We merged the answer to this comment with the one to comment 2 Referee #2.

(14) P4 Line 10: I would remove "seasonal" as you sampled only 7 months of the year. Furthermore, bacterial concentration in May 2014 is higher or equivalent to those in July/August.

"Seasonal" was removed from the sentence.

**(15) P5 Line 16: Replace "is" with "in"**

This sentence now reads as: "...increase more substantially above background values at high wind speed, than in the case of bacterial cells".

**References mentioned in the reply**

IAEA International Atomic Energy Agency: Environmental isotopes in the hydrological cycle. Principles and applications. Vol. 2. Atmospheric Water, available at: http://www-naweb.iaea.org/napc/ih/IHS\_resources\_publication\_hydroCycle\_en.html (last access on 30.01.2017), 2001.

Lloyd, G., Choularton, T. W., Bower, K. N., Gallagher, M. W., Connolly, P. J., Flynn, M., Farrington, R., Crosier, J., Schlenczek, O., Fugal, J., and Henneberger, J.: The origins of ice crystals measured in mixed-phase clouds at the high-alpine site Jungfraujoch, Atmos. Chem. Phys., 15, 12953-12969, doi:10.5194/acp-15-12953-2015, 2015.

Stopelli, E., Conen, F., Zimmermann, L., Alewell, C., and Morris, C. E.: Freezing nucleation apparatus puts new slant on study of biological ice nucleators in precipitation, Atmos. Meas. Tech., 7, 129-134, doi:10.5194/amt-7-129-2014, 2014.

Stopelli, E., Conen, F., Morris, C. E., Herrmann, E., Bukowiecki, N., and Alewell, C.: Ice nucleation active particles are efficiently removed by precipitating clouds, Sci. Rep., 5, 16433, doi:10.1038/srep16433, 2015.

Stopelli, E., Conen, F., Morris, C. E., Herrmann, E., Henne, S., Steinbacher, M., and Alewell, C.: Predicting abundance and variability of ice nucleating particles in precipitation at the high-altitude observatory Jungfraujoch, Atmos. Chem. Phys., 16, 8341-8351, doi:10.5194/acp-16-8341-2016, 2016.

**Referee #2**

**General comments**

This study represents the results of INP measurements and bacteria cell counting of 56 snow samples collected at Jungfraujoch over an 18-month sampling period. They found a larger dynamic range of INP active at -8° C compared to the number of bacterial cells and a high fraction of living bacteria cells. Correlations with water vapor loss prior sampling show that INP have shorter atmospheric residence times than bacterial cells, in particular at low wind speeds. Furthermore, 24 strains of Pseudomonas syringae were isolated from selected samples and phylogenetically characterized. The manuscript is well written and the results are clear and well presented.

Specific comments:

(1) I suggest renaming the section 3 from "results" to "results and discussion" as there is no separate discussion section.

**Done**

(2) When discussing the higher concentration dynamics of the INP compared to bacteria the authors mention changing source and sink strength of INP and bring a one short statement about "other particles carrying a biological INP" on P4 Line20. This discussion seems very general and should be extended. Other possible biological INP sources (at least fungi) should be specifically mentioned and more references should be cited. This would also help to explain the INP-8 concentrations at JFJ as P. syringae was found only in three out of 13 tested samples. Given the high diversity of P. syringae strains as found in those samples and an average of 60% of total living bacteria cells in the samples one could actually expect to find P. syringae in more samples. Preferential removal of P. syringae with precipitation and loss of culturability during atmospheric transport as stated in the MS seems possible, but possible sources of non-P. syringae INP active at -8°C should be discussed in more detail.

Also in the light of comment 13 Referee #1, we changed the paragraph involving page 4 line 20, which now read as: "Hence, the relatively small DR of bacterial cells compared to that of  $INPs_{-8}$  suggests that the majority of bacterial cells has a longer residence time in the atmosphere than  $INPs_{-8}$ . The shorter residence time of  $INPs_{-8}$  is likely to be due to their capacity to catalyse ice formation and growth at -8 °C, leading to their rapid deposition with the growing crystal. Such  $INPs_{-8}$ , include not only bacterial cells, but also fungal spores, pollen, parts thereof, and soil particles associated with biological ice nucleation active fragments (Conen et al., 2011; Fröhlich-Nowoisky et al., 2015; Hill et al., 2016; Joly et al., 2013; Morris et al., 2013b; O'Sullivan et al., 2015, Pummer et al., 2011; Fröhlich-Nowoisky et al., 2015; Hill et al., 2016; Joly et al., 2015; Hill et al., 2016 and Morris et al., 2013b.

(3) Some more information about the sampling conditions would be helpful as the station at JFJ is not always within clouds. Did the authors only collect precipitation/snow when the station was within clouds? And if not, did they found differences in the number of INP and bacterial cells in samples collected in a cloud compared to samples collected below a cloud? Why was there no sampling in winter?

The Station was always inside precipitating clouds while sampling, in order to reduce the influence of scavenging of particles on our measurements. At page 2 line 19 this sentence has been added: "The Station was always inside precipitating clouds while collecting snow". Concerning sampling in winter, please refer to our reply to comment 12 Referee #1.

**Other comments/typos:**

(4) P2 Line 23: Although the math is correct here, I think it is better to consistently use either 0.9 % (as on P3 Line 18) or 9 ‰ NaCl.

Changed into 9 m for consistency, also in the Supplement

**(5) P2 Line 23: Please provide some basic information here. How many droplets did you measure? What was the volume of the droplets? What kind of controls were measured?**

Also according to comment 8 Referee #1, this paragraph has been changed into: "Snow samples were melted at room temperature (about 16 °C) and adjusted to physiological conditions (9 ‰ NaCl) to prevent osmotic stress on cells and improve the detection of freezing events. They were analysed immediately on site for concentration of INPs active between -2 and -12 °C in a drop freeze assay with the LINDA device loaded with 52 Eppendorf Safelock tubes containing 100  $\mu$ L of sample each, as described in Stopelli et al. (2014). For each sample, two filters were prepared for later analysis of bacterial number concentration. The reference to controls has been introduced at page 2 line 32: "Blanks for the determination of INPs and bacterial cells were periodically prepared using the Milli-Q water used to rinse the tin as control sample."

(6) Table 1 caption: I suggest to change the beginning to "Diversity of *Pseudomonas syringae* strains" Done

**References mentioned in the reply**

Conen, F., Morris, C. E., Leifeld, J., Yakutin, M. V., and Alewell, C.: Biological residues define the ice nucleation properties of soil dust, Atmos. Chem. Phys., 11, 9643–9648, doi:10.5194/acp-11-9643-2011, 2011.

Fröhlich-Nowoisky, J., Hill, T. C. J., Pummer, B. G., Yordanova, P., Franc, G. D., and Pöschl, U.: Ice nucleation activity in the widespread soil fungus *Mortierella alpina*, Biogeosciences, 12, 1057–1071, doi:10.5194/bg-12-1057-2015, 2015.

Hill, T. C. J., DeMott, P. J., Tobo, Y., Fröhlich-Nowoisky, J., Moffett, B. F., Franc, G. D., and Kreidenweis, S. M.: Sources of organic ice nucleating particles in soils, Atmos. Chem. Phys., 16, 7195-7211, doi:10.5194/acp-16-7195-2016, 2016.

Morris, C. E., Sands, D. C., Glaux, C., Samsatly, J., Asaad, S., Moukahel, A. R., Goncalves, F. I. T., and Bigg, K. E.: Urediospores of rust fungi are ice nucleation active at > -10 °C and harbor ice nucleation active bacteria, Atmos. Chem. Phys. 13, 4223-4233, doi:10.5194/acp-13-4223-2013, 2013.

**Additional insertions from the authors of the manuscript**

- *Cts* gene sequences have been in the meanwhile deposited in the online repository GenBank. Therefore we added at the end of section 2.2 the sentence: "Reported sequences are deposited in the GenBank Archive under the accession numbers GenBank KY379248-KY379271." As a consequence, we remove the provisory Excel table reporting such sequences which was previously sent a supplementary material to the article.
- "Colony forming units" changed into "colony-forming units" through the text
- Page 1 line 13 sentence in brackets changed into: "(of which 60 % was on average alive)"
- Page 1 line 17 changed "perhaps" into "likely"
- Page 1 line 23 changed "plants" into "habitats", now reads as: "and its propensity to spread to colonise new habitats"
- Page 1 line 32 we removed "the spread of microbial"
- Page 2 line 2, added "a", now reads as: "corresponding to a cold temperature regime and supersaturation"
- Page 2 line 28 substituted "live" with "alive"
- Page 2 line 31 added "the" to "in the dark"

- Page 3 line 2 substituted "to study" with "to assess"
- Page 3 line 7 substituted "successive" with "subsequent"
- Page 3 line 11 added "of KBC", now reads as: "without the antibiotics and boric acid of KBC"
- Page 3 line 14 added "Research", now reads as: "Plant Pathology Research Unit", and removed "the" from "for molecular identification"
- Page 3 line 22 modified "tobacum" into "tabacum"
- Page 3 line 30 modified the beginning of the sentence into "Analysis of partial..."
- Page 3 line 38 modified into "and data were stored"
- Title of Paragraph 3.1 now reads as: "Variability of the concentrations of bacterial cells and INPs 8"
- Page 4 line 23 added "a" to the sentence "with a small standard deviation"
- Page 5 line 3 sentence modified into: "This opens the question whether we can find evidence also for the replenishment of their atmospheric concentrations at shorter time scales."
- Page 5 lines 12/13 modified "is" into "would be" and removed "it is", sentence now reads as: "Therefore, their background number concentration would be relatively large and stable and not changed much by the additional cells aerosolised at high wind speeds in the region."
- Page 6 line 23/24 beginning of the sentence modified into: "The decrease of INPs relative to bacterial cells with precipitation..."
- Page 6 line 28 added "a" to "P. syringae appeared as a minor component"
- Page 6 line 31 added "...and new habitats."
- Figure 1 and 2: units in the y-axes have been inserted in brackets
- Page 7 lines 5 and 6 now read as: "Caroline Guilbaud provided fundamental support in the isolation of *P. syringae* and did the phylogenetic analysis on the isolated strains."
- Page 7 line 13 modified into: "Urs and Maria Otz, as well as Joan and Martin Fisher are acknowledged for their support on site."
- Page 13 lines 4/5: substituted "where" with "with" and "for" with "with", this sentence now reads as: "Red symbols correspond to values with average wind speeds > 50 km h-1 during samplings and blue symbols correspond to data with lower wind speeds."
- In the Supplement changed "Supplement of" into "Supplement to"
- In Supplemental Table 1, column *Pseudomonas syringae*, substituted "under 2" with " $\leq 2$ "

**Interpretation at Jungfraujoch 1 Ice nucleators, bacterial cells and Pseudomonas syringae in 2 precipitation at Jungfraujoch**

Emiliano Stopelli1, Franz Conen1, Caroline Guilbaud2, Jakob Zopfi3, Christine Alewell1,
Cindy E. Morris2

5 1Environmental Geosciences, University of Basel, 4056 Basel, Switzerland

6 2INRA PACA, UR 0407 Plant Pathology Research Unit, 84143 Montfavet, France

7 3Acquatic and Stable Isotope Biogeochemistry, University of Basel, 4056 Basel, Switzerland

8 Correspondence to: Emiliano Stopelli (emiliano.stopelli@unibas.ch) and Franz Conen (franz.conen@unibas.ch)

9

10 Abstract Ice nucleation is a means by which the deposition of an airborne microorganism can be accelerated under favourable meteorological conditions. Analysis of 56 snow samples collected at the high altitude 11 12 observatory Jungfraujoch (3580 m a.s.l.) revealed an order of magnitude larger dynamic range of ice nucleating 13 particles active at -8 °C (INPs.8) compared to the total number of bacterial cells (of which 60 % was on average 14 alive). This indicates a shorter atmospheric residence time for INPs.8. Furthermore, concentrations of INPs.8 15 decreased much faster, with an increasing fraction of water precipitated from the air mass prior to sampling, than the number of total bacterial cells. Nevertheless, at high wind speeds (> 50 km h-1) the ratio of INPs.8 to total 16 bacterial cells largely remained in a range between  $10^{-2}$  to  $10^{-3}$ , independent of prior precipitation, Jikely because 17 18 of recent injections of particles in regions upwind. Based on our field observations, we conclude that ice 19 nucleators travel shorter legs of distance with the atmospheric water cycle than the majority of bacterial cells. A 20 prominent ice nucleating bacterium, Pseudomonas syringae, has been previously supposed to benefit from this 21 behaviour as a means to spread via the atmosphere and to colonise new host plants. Therefore, we targeted this 22 bacterium with a selective cultivation approach, P. syringae was successfully isolated for the first time at such an 23 altitude in 3 of 13 samples analysed. Colony forming units of this species constituted a minor fraction ( $10^{-4}$ ) of 24 the numbers of INPs.8 in these samples. Overall, our findings expand the geographic range of habitats where this 25 bacterium has been found and corroborates theories on its robustness in the atmosphere and its propensity to 26 spread to colonise new habitats.

**27 1 Introduction**

28 The nucleation of ice in clouds is a process of primary relevance both for the radiative budget of clouds and for 29 the development of precipitation (Cantrell and Heymsfield, 2005; Mülmenstädt et al., 2015; Murray et al., 2012). Most ice nucleating particles (INPs) active at moderate supercooling in the atmosphere are of biological origin 30 31 (Murray et al., 2012). Pseudomonas syringae was the first organism found to produce an ice nucleation active 32 molecule (Maki et al., 1974) and to occur in clouds as potential biological INP (Sands et al., 1982). As it is also a 33 plant pathogen it received and continues to receive attention from biologists in the perspective of improving the 34 protection of crops from diseases (Lamichhane et al., 2014; Lamichhane et al., 2015). The combination of both 35 roles, of an ice nucleator and of a plant epiphyte and pathogen, sparked the bioprecipitation hypothesis (Morris et 36 al., 2014; Sands et al., 1982). Part of the hypothesis is the idea that ice nucleation activity contributes 37 preferentially to the deposition of the organisms with this property helping them to return to plant surfaces where

Deleted: was previously Formatted: Font: Not Italic Deleted: In this sense Deleted: Deleted: isolated Formatted: Font: Italic Deleted: it for the first time at such an altitude from 3 of 13 samples analysed Deleted: Pseudomonas syringae, a prominent ice nucleating bacterium, was successfully isolated for the first time at such an altitude from 3 of 13 samples analysed. Deleted: Deleted: and Deleted: plants

| -      | Deleted: an                      |
|--------|----------------------------------|
| $\neg$ | Deleted: active                  |
| Υ      | Deleted: in clouds               |
| {      | Deleted: the spread of microbial |

[revised manuscript text omitted]
 Meteoral Soc                           | Polotodi Criffitha A.D. Carran E         |
| n, w. and reginstread, A.: Houdetion of ice in hoposphere clouds. A Review, D. Am. Meteorol. Soc.,                     | Weingartner, E., Zimmermann, L.,         |
| E Morris C E Laifeld I Vakutin M V and Alawall C: Biological residues define the ice                                   | Chambers, S. D., Williams, A. G., and    |
| y r., Montis, C. E., Leffeld, J., Takutili, M. V., and Aleweir, C.: Diological residues define the re-                 | Steinbacher, M.: Surface-to-mountaintop  |
| ucceation properties of soil dust, Atmos. Chem. Phys., 11, 9045–9048, doi:10.5194/acp-11-9045-2011,                    | observations at the Jungfraujoch, Atmos. |
| 2011.
24 Normaisland I. Hill T. C. I. Durannan, D. C. Mandanarus, D. Farna, C. D. and Disable H. Las musication.    | Chem. Phys., 14, 12763-12779,            |
| ch-Nowoisky, J., Hill, T. C. J., Pummer, B. G., Yordanova, P., Franc, G. D., and Poschi, U.: Ice nucleation            | doi:10.5194/acp-14-12/63-2014, 2014.     |
| ctivity in the widespread soil fungus Mortierella alpina , Biogeosciences, 12, 1057–1071, doi:10.5194/bg-       |                                          |
| 2-1057-2015, 2015.                                                                                              |                                          |
| ud, C., Morris, C. E., Barakat, M., Ortet, P., Berge, O.: Isolation and identification of Pseudomonas           |                                          |
| yringae facilitated by a PCR targeting the whole P. syringae group, FEMS Microbiol. Ecol., 92,                         |                                          |
| loi:10.1093/temsec/tiv146, 2016.                                                                                       |                                          |
| er, Ø., Harper, D. A., and Ryan, P. D.: PAST: PAleontologi- cal STatistics software package for education              | Formatted: Indent: Left: 0 cm,           |
| and data analysis, Palaeontol. Electron., 4, available at: http://palaeo-                                              | Hanging: 0.85 cm                         |
| electronica.org/2001_1/past/issue1_01.htm (last access: 14.11.2016), 2001.                                             |                                          |
| C. J., DeMott, P. J., Tobo, Y., Fröhlich-Nowoisky, J., Moffett, B. F., Franc, G. D., and Kreidenweis, S.               |                                          |
| A.: Sources of organic ice nucleating particles in soils, Atmos. Chem. Phys., 16, 7195-7211,                           |                                          |
| loi:10.5194/acp-16-7195-2016, 2016.                                                                                    |                                          |
| International Atomic Energy Agency: Environmental isotopes in the hydrological cycle. Principles and                   | Formatted: English (U.S.)                |
| pplications. Vol. 2. Atmospheric Water, available at: http://www-                                                      |                                          |
| aweb.iaea.org/napc/ih/IHS_resources_publication_hydroCycle_en.html (last access on 30.01.2017), 2001.                  | Formatted: English (U.S.)                |
| A., Attard, E., Sancelme, M., Deguillaume, L., Guilbaud, C., Morris, C. E., Amato, P., and Delort, A. M.:              | Formatted: English (U.S.)                |
| ce nucleation activity of bacteria isolated from cloud water, Atmos. Environ., 70, 392-400,                            |                                          |
| loi:10.1016/j.atmosenv.2013.01.027, 2013.                                                                              |                                          |
| M., Amato, P., Deguillaume, L., Monier, M., Hoose, C., and Delort, A. M.: Quantification of ice nuclei                 |                                          |
| active at near 0 °C temperatures in low-altitude clouds at the Puy de Dôme atmospheric station, Atmos.                 |                                          |
| Chem. Phys., 14, 8185–8195, doi:10.5194/acp-14-8185-2014, 2014.                                                        |                                          |
| hhane, J. R., Varvaro, L., Audergon, J. M., Parisi, L., and Morris, C. E.: Disease and frost damage of                 |                                          |
| woody plants caused by Pseudomonas syringe : seeing the forest for the trees, in: Advances in Agronomy , |                                          |
| park, D. L., ed., Academic Press, pp. 235-295, 2014.                                                                   |                                          |
| hhane, J. R., Messean, A., and Morris, C. E.: Insights into epidemiology and control of diseases of annual             |                                          |
| plants caused by the Pseudomonas syringae species complex, J. Gen. Plant Pathol., 81, 331-350,                         |                                          |
| loi:10.1007/s10327-015-0605-z, 2015.                                                                                   |                                          |
| nann, J. and Upper, C. D.: Aerial dispersal of epiphytic bacteria over bean plants. Appl. Environ.                     |                                          |
|                                                                                                                        |                                          |

Bauer, H., Kasper-Giebl, A., and content of cloud water doi:10.1016/S0169-8095(02 Berge, O., Monteil, C. L., Bartoli, guide to a data base of the d this phylogenetic complex, P Burrows, S. M., Butler, T., Jöckel, global atmosphere - Part 2: Chem. Phys., 9, 9281-9297, Cantrell, W. and Heymsfield, A .: 86, 795-807, doi:10.1175/BA Conen, F., Morris, C. E., Leifeld nucleation properties of soil 2011. Fröhlich-Nowoisky, J., Hill, T. C. activity in the widespread soil 12-1057-2015, 2015. Guilbaud, C., Morris, C. E., Bara syringae facilitated by a P doi:10.1093/femsec/fiv146, 2 Hammer, Ø., Harper, D. A., and R and data analysis, electronica.org/2001\_1/past/ Hill, T. C. J., DeMott, P. J., Tobo, M.: Sources of organic i doi:10.5194/acp-16-7195-20 **IAEA** International Atomic Energy applications. Vol. naweb.iaea.org/napc/ih/IHS\_ Joly, M., Attard, E., Sancelme, M. Ice nucleation activity of doi:10.1016/j.atmosenv.2013

Amato, P., Parazols, M., Sancelme, M., Laj, P., Mailhot, G., and Delort, A. M.: Microorganisms isolated from

- 40 Joly, M., Amato, P., Deguillaume 41 active at near 0 °C temperat 42 Chem. Phys., 14, 8185-8195
- 43 Lamichhane, J. R., Varvaro, L., A 44 woody plants caused by Psei 45 Spark, D. L., ed., Academic I
- 46 Lamichhane, J. R., Messean, A., an 47 plants caused by the Pseul 48 doi:10.1007/s10327-015-060
- 49 Lindemann, J. and Upper, C. D. 50 Microbiol., 50, 1229-1232, 1985.
- 51 Maki, L. R., Galyan, E. L., and Caldwell, D. R.: Ice Nucleation Induced by Pseudomonas syringae, Appl. 52 Environ. Microb., 28, 456-459, 1974.

[revised manuscript text omitted]